# Community structure and niche differentiation of endosphere bacterial microbiome in *Camellia oleifera*

Yan Zhang,[1] Chu Ting Ding,[2] Taoya Jiang,[2] Yu Hua Liu,[2] Yang Wu,[1] Hui Wen Zhou,[1] Li Sha Zhang,[1] Ye Chen[2]

**ABSTRACT** Understanding the changes in bacterial community structure in different microenvironments of *Camellia oleifera* is essential to better explore the benign interaction between beneficial microorganisms and plants. Using *Camellia oleifera* trees, a Chinese wooden oil plant as a model ecosystem, we characterized the archaeal and bacterial microbiome across five different tissue-level niches using 16S rRNA gene analyses. Our research indicates that the diversity of *Camellia oleifera* endophytic bacterial communities is highly dependent on the plant compartment. The species replacement process (69.90%) is the dominant factor in the differences in bacterial community structure. The dominant bacteria phyla (*Proteobacteria*, *Acidobacteria*, *Actinobacteria*, *Bacteroidetes*, *Firmicutes*, *Chloroflexi*, and *Verrucomicrobia*) of *Camellia oleifera* show a significant plant compartment (roots, stems, leaves, fruits) enrichment effects. A variety of bacteria (*Hymenobacter*, *Allorhizobium-Neorhizobium-Pararhizobium-Rhizobium*, *Mesorhizobium*, *Bradyrhizobium*, *Bacillus*, *Ochrobactrum*, *Pantoea*, *Pseudomonas*, etc.) with nitrogen-fixed potentials are enriched in *Camellia oleifera* tissue. In addition, the hub bacterial groups of *Camellia oleifera* are *Nitrospira*, *Haemophilus*, *Staphylococcus*, *Ruminiclostridium*, and *Ochrobactrum*. They are widespread colonization in various tissues with a low relative abundance and may play an important role in the nitrogen cycle, host life promotion, and plant defense. This study provides a holistic understanding of the endosphere bacterial community structure, which is one of the most complete ecological niche-level analyses of *Camellia oleifera*. These results provide a scientific theoretical basis for an in-depth discussion of plant-endosphere microbial interaction and better exploration of benign interaction of beneficial microorganisms and plants.

**IMPORTANCE** Microorganisms inhabited various tissues of plants and play a key role in promoting plant growth, nutritional absorption, and resistance. Our research indicates that the diversity of *Camellia oleifera* endophytic bacterial communities is highly dependent on the plant compartment. *Proteobacteria*, *Acidobacteria*, *Actinobacteria*, *Bacteroidetes*, *Firmicutes*, *Chloroflexi*, and *Verrucomicrobia* are dominant bacteria phyla. The tissues of *Camellia oleifera* contain various bacteria with nitrogen fixation potential, host life promotion, and plant defense. This study provides a scientific theoretical basis for an in-depth discussion of plant-endosphere microbial interaction and better exploration of benign interaction of beneficial microorganisms and plants.

**KEYWORDS** bacterial community, ecology niche, endosphere, *Camellia oleifera*

*Camellia oleifera* is a unique oil plant with high nutritional value in China and one of the world's four major wooden oil plants. Its uses have high economic flexibility, such as the production of Camellia seed oil, cosmetics, chemicals, brachycal, gums, activated carbon, and the cultivation of edible fungi (1). In addition, *Camellia oleifera* are basically grown in mountains and hill regions that are not suitable for grain production,

Address correspondence to Yan Zhang, zhangyan0856@hotmail.com.

The authors declare no conflict of interest.

See the funding table on p. 14.

thereby avoiding competing for land resources with grain. However, the long-term predatory and extensive management have led to a significant decrease in soil nutrients and in a low-yield state.

Plants are colonized by complicated multi-kingdom microbial communities (bacteria, fungi, native creatures, etc.) (2, 3). Each part of the plant is a unique ecosystem of microorganisms. Compared with other plant tissues (including roots, stems, leaves, flowers, and seeds), it has a unique microbial assembly (4–7). The results of Xiong et al. (8) pointed out that some members of *Burkholderiaceae*, *Microbacteriaceae*, *Streptomycetaceae*, and *Rhizobiaceae* were enriched on the surface of phylloplane and rhizoplane at the maize seedling stage. Lei et al. (9) show that the bacterial community structure between different parts of *Macleaya cordata* has a significant change, among which *Sphingomonas* and *Methylobacterium* dominate the fruits and leaves, respectively. In addition, a significant plant compartment effect was observed in the microbiome of tomato, willow, poplar, agave, and cactus (10–13). In addition, only adaptive or non-picky bacterial populations can survive or flourish within plant tissues due to filtration and selection, which leads to a low degree of microbial diversity (14).

These microorganisms inhabiting various parts of plants can play a key role in promoting plant growth, nutrient absorption, and resistance to biotic or abiotic stresses (diseases, insect pests, high temperature, saline-alkali soil or drought, etc.) (15–17). One of the strategies developed by non-legume plants to increase the supply of nitrogen is to form a nutritional alliance with endophyte nitrogen fixation bacteria. So far, a large number of nitrogen-fixed nutrient bacteria (*Azospirrillum brasiliense*, *Gluconacetobacter diazotrophicus*, *Herbaspirillum seropedicae*, *Azoarcus,* etc.) have been identified as epibiotic or endophyte bacteria, combined with cereal and grass (18). Recently, Deynze et al. (19) reported that they identified landrace maize that could benefit from the atmospheric nitrogen fixed by the related endophytic nitrogen-fixing bacteria (*Azospirillum brasilense*, *Herbaspirillum seropodicae*, and *Burkholderia unamae*). The aerial root mucus produced by this special maize is proved to be the environmental niche of the *nif* gene pool, which can provide up to 85% of assimilated nitrogen. Under drought stress, *Bacillus sp*. (12D6) and *Enterobacter sp*. (16i) will rapidly colonize in the rhizosphere of maize seedlings, stimulate the secretion of auxins and gibberellins, significantly increase the root length, root surface area, and number of root tips of maize seedlings, to obtain more water and alleviate drought stress (20). Therefore, analyzing the characteristics of bacterial communities in different ecological niches of healthy hosts may help to improve soil quality, crop growth, and stress resistance, thus reducing the dependence on fertilizers in production activities. It is of great importance for promoting the sustainable development of *Camellia oleifera* production and understanding the contribution of *Camellia oleifera* to ecosystem services.

Most of the previous research is concentrated in the ecological position of the bacterial community of the soil-root interface (21–23). Contrary to the knowledge of bacterial microbial community differentiation related to the rhizosphere (24–26), there are few reports on the structural composition of bacterial communities in different tissues of *Camellia oleifera*, especially the relationship between underground and aboveground communities. Here, we employed 16S rRNA sequencing to evaluate the niche differentiation of bacterial communities related to bulk soil and roots, stems, leaves, and fruits of *Camellia oleifera*. The analysis of niche differentiation (bulk soil, root, stem, leaf, and fruit) of endophytic bacteria community in *Camellia oleifera* can provide a scientific basis for further exploring the mechanism of plant endophytic microbial interaction and tapping the biological potential of benign interaction between plant growth and development and beneficial microorganisms.

## MATERIALS AND METHODS

### Study location and sampling methods

The *Camellia oleifera* tree (planted for about 10 years) planted in Hualong New Village, Yongxiu County, Jiujiang City was selected to obtain samples for this study. The forest land was established in April 2012. The planting density of *Camellia oleifera* is 1,100–1,300 trees per hectare, and the row spacing is 3.0 m. The samples were collected on 12 April 2021. At the time of sampling, the average height of *Camellia oleifera* was about 2.0 m. Six healthy individuals with basically the same growth vigor were selected, and the samples collected included bulk soil, roots, stems, leaves, and fruits. Samples of bulk soil and roots were collected at a depth of 5–20 cm below ground level. Sterile tools were used to take out the root, and the soil that is not tightly attached was removed by shaking. The roots were cut into 2–3 cm fragments, put it in a tube equipped with PBS buffer and glass beads, and the vortex was washed to obtain the root sample. Bulk soil samples were collected at a distance of 30–50 cm from the main stem using a sterile soil drill. For leaf, stem, and fruit samples, a fruiting branch was collected from six *Camellia oleifera* individuals. The average circumference of the sampled branches is about 1 cm and the height is about 120 cm. 10 subsmall stems with bark were collected from each plant as stem samples. All leaves and fruits collected from the sampled branches are called leaf and fruit samples. Weeds were controlled manually in each autumn. No obvious diseases or pest damage heve been observed in recent years. Irrigation was not applied throughout the plant period.

### Sample preparation

The samples were processed as described by Beckers et al. (11). The soil particles that are tightly attached to the root surface are removed by shaking on the oscillator (20 min, 120 rpm). Subsequently, the root, stem, leaf, and fruit samples were sterilized by 75% (vol/vol) ethanol (2 min), NaClO (2.5% active Cl$^-$ and 0.1% Tween 80) (5 min), 70% (vol/vol) ethanol (30 s), and sterile ultrapure water (washing the samples for five times). A sterile scalpel was used to divide the plant sample into small segments, and a Polytron PR1200 mixer (Kinematica A6) was used to soak it in sterile phosphate-buffered saline (PBS; 130 mM NaCl, 7 mM Na$_2$HPO$_4$, 3 mM NaH$_2$PO$_4$, pH 7.4). Sterilization and homogenization of plant samples are carried out under sterile conditions. Finally, each sample of all homogeneous plant materials (roots, stems, leaves, or fruits) is stored at −80 °C until DNA is extracted.

### DNA extraction, PCR amplification, and 16S rRNA sequencing

Soil DNA was extracted using the E.Z.N.A. Soil DNA Kit (Omega Bio-tek, Norcross, GA, USA) according to the manufacturer's instructions. For plant tissue, the first equal portions of homogeneous plant material (1.5 mL) (13,400 rpm, 30 min) were centrifuged to collect all cells. The supernatant was discarded and DNA from the precipitated plant materials was extracted. According to the manufacturer's scheme (Strategic Biomedical AG, Birkenfeld, Germany), DNA was extracted from plant samples using the Invisorb Spin Plant Mini Kit. The V3–V4 hypervariable regions of the bacterial 16S rRNA gene were amplified by PCR using a special primer pair (335F: 5′-CAD ACT CCT ACG GGA GGC-3′, and 769R: 5′-ATC CTG TTT GMT MCC VCV RC-3′), where the barcode comprised a six-base sequence that was unique to each sample. PCR was performed using a protocol similar to the method described by Zheng et al. (27). 16S rRNA amplicons were pooled and then sequenced with Illumina Hiseq 2500 (Biomarker Technologies, Beijing, China). The sequences obtained in this study were deposited in the Genome Sequence Archive (GSA) of the National Genomics Data Center, under accession number CRA009115.

### Illumina Hiseq sequence processing

Raw sequence data were quality filtered: (i) Trimmatic (version 0.33) was used to filter the quality of raw sequence data and remove the reads whose average quality value is

lower than 20 (28); (ii) Cutadapt (version 1.9.1) was used to identify and remove primer sequences (29); and (iii) USEARCH (version 10) was utilized to conduct chimera detection and operational taxonomic units (OTUs) clustering (97% similarity) (30). Taxonomy was identified for each OTU using the RDP classifier (31) trained on the Greengenes (32) and Silva (33) databases for bacterial sequences. In this study, after the quality control of the raw sequence, 1,893,703 clean reads were generated, with an average length of 432 bp. The number of clean reads per sample ranged from 46,616 to 72,362. After the data were standardized, QIIME2 was used to calculate the richness and diversity index (ACE, Chao 1, Simpson, and Shannon Weaver). The rarefaction curve and Shannon curve of bacteria indicate that our sequencing data represent most of the composition (Fig. S1).

## Statistical analysis

Whether the distribution of parameters conforms to normal distribution, ANOVA or Kruskal-Wallis rank sum test is used to evaluate the significant difference of variance of parameters (ACE, Chao 1, Simpson, and Shannon index). When $P < 0.05$, post hoc comparisons were employed by either Tukey's honest significant differences tests or pairwise Wilcoxon rank-sum tests.

Based on the Bray-Curtis distance of OTU abundance, community compositional dissimilarities were estimated. To examine the elevational differences in compositional dissimilarities, principal coordinate analysis (PCoA) and permutational multivariate analysis of variance (PERMANOVA) were performed in R package vegan. Compositional dissimilarities among plant compartments (β-diversity) were partitioned into replacement (Podani family) and richness difference components (Sørensen dissimilarities) using the R package adespatial (34).

EdgeR (R3.3.0) was used to identify the differential expression of bacteria between the two niches, and the genus with a possibility value (*P* value) higher than 0.05 was defined as the differential genus. Based on Spearman's correlation coefficient (Spearman's $|r|$ > 0.7 or $P < 0.05$), MCODE in Cytoscape (v.3.8.2) was used to analyze the network at the bacterial genus level (35), and CytohHubba was used to analyze the core bacterial community of *Camellia oleifera* (36). The networks were visualized in Cytoscape (v.3.8.2). Functional annotations of prokaryotic taxa were carried out using FAPROTAX (v.1.1).

## RESULTS

### α-diversity of the bacterial community in each plant compartment

The α-diversity of the bacterial community in plant compartments of *Camellia oleifera* is lower than that of bulk soil (Fig. 1; Table S1). The bacterial richness (ACE and Chao1 index) and diversity (Simpson and Shannon index) of bulk soil (S) were significantly higher than fruit endosphere (FE), leaf endosphere (LE), and stem endosphere (SE), and slightly higher than root endosphere (RE). The richness and diversity of endophytic bacteria in LE were significantly higher than those in SE and FE. Compared with SE, the diversity index (Simpson: 0.947 ± 0.02, Shannon: 5.82 ± 0.8) was significantly reduced under FE, but the richness (ACE and Chao1 index) showed no significant differences between SE and FE.

### β-diversity of the bacterial community in each plant compartment

PCoA based on OTU level shows that there are significant differences in bacterial communities and compositions of different niches of *Camellia oleifera* (Fig. 2 and Table 1). The analysis using Bray-Curtis distance showed that there were different patterns of bacterial communities related to the two axes, which explained 37.52% and 17.78% of the total variation in each compartment of *Camellia oleifera*.

PERMANOVA analysis (Bray-Curtis distance) shows that the composition of bacterial communities in different niches is significantly different (Table 1). Compared with bulk soil, the composition of the bacterial community in plant compartments is significantly different, forming an independent individual group (FE and S, $R^2$ = 0.6309, $P < 0.01$; LE

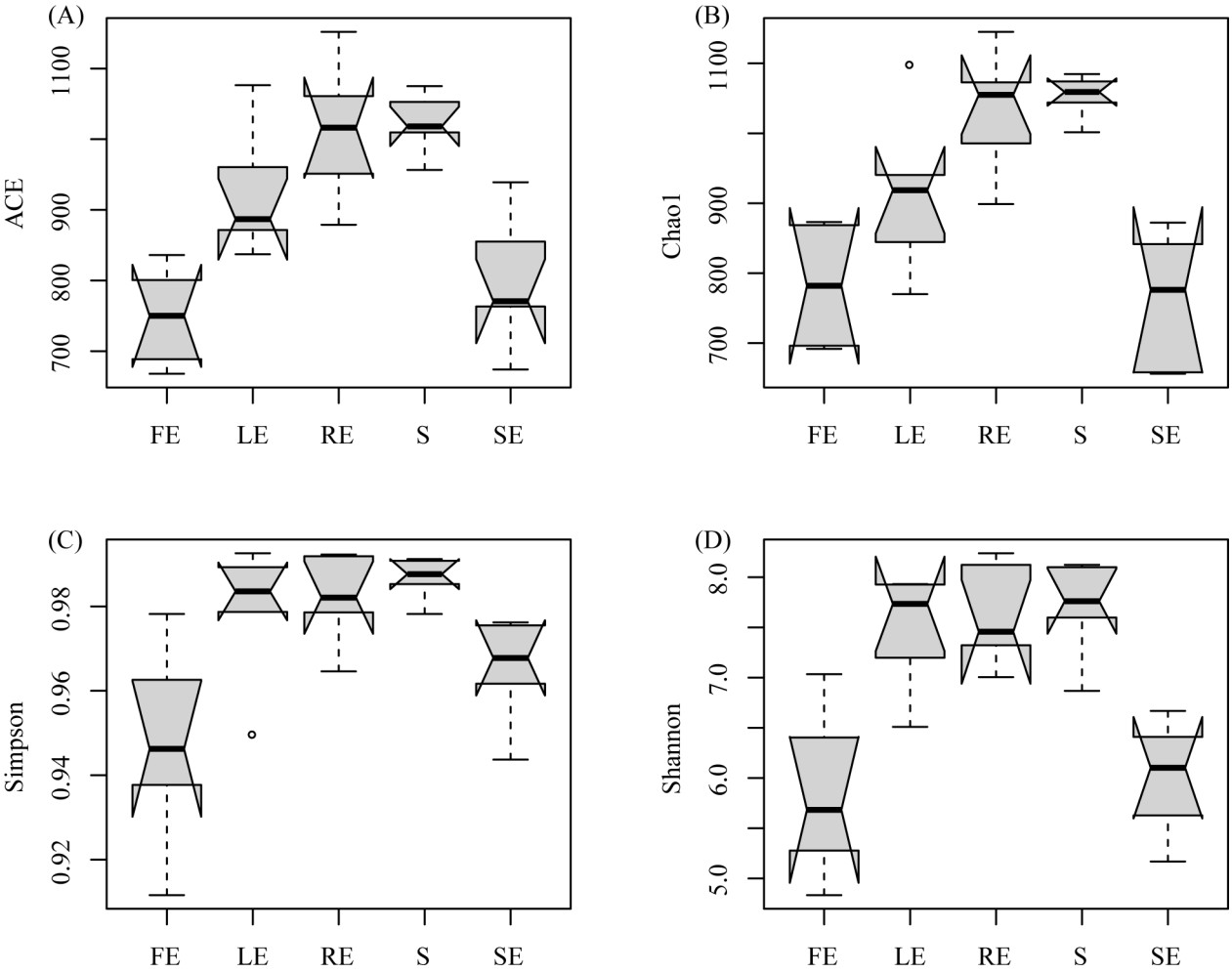

**FIG 1** The α-diversity within each plant compartment of *Camellia oleifera*. (A). ACE, (B) chao1, (C) Simpson and (D) Shannon. The FE, LE, SE, RE, and S represent the fruit endosphere, leaf endosphere, stem endosphere, root endosphere, and bulk soil, respectively.

and S, $R^2 = 0.4449$, $P < 0.01$; RE and S, $R^2 = 0.4335$, $P < 0.01$; SE and S, $R^2 = 0.6262$, $P < 0.001$). Except for LE and RE, the community structure of endophytic bacteria in different compartments (FE and LE, $R^2 = 0.3987$, $P < 0.01$; FE and RE, $R^2 = 0.4902$, $P < 0.01$; FE and

**TABLE 1** Permutational multivariate ANOVA results with Bray-Curtis distance matrices implemented to determine the composition of the bacterial community in different niches of *Camellia oleifera* was significantly different[a]

| Pairs | $R^2$ | P value | P adjusted |
|---|---|---|---|
| FE vs LE | 0.398655 | 0.003 | 0.0063 |
| FE vs RE | 0.490232 | 0.002 | 0.0063 |
| FE vs S | 0.630864 | 0.005 | 0.0063 |
| FE vs SE | 0.171071 | 0.017 | 0.0189 |
| LE vs RE | 0.093058 | 0.389 | 0.3890 |
| LE vs S | 0.444895 | 0.004 | 0.0063 |
| LE vs SE | 0.423546 | 0.004 | 0.0063 |
| RE vs S | 0.433534 | 0.005 | 0.0063 |
| RE vs SE | 0.501711 | 0.001 | 0.0050 |
| S vs SE | 0.62617 | 0.001 | 0.0050 |

[a]The FE, LE, SE, RE, and S represent the fruit endosphere, leaf endosphere, stem endosphere, root endosphere, and bulk soil, respectively.

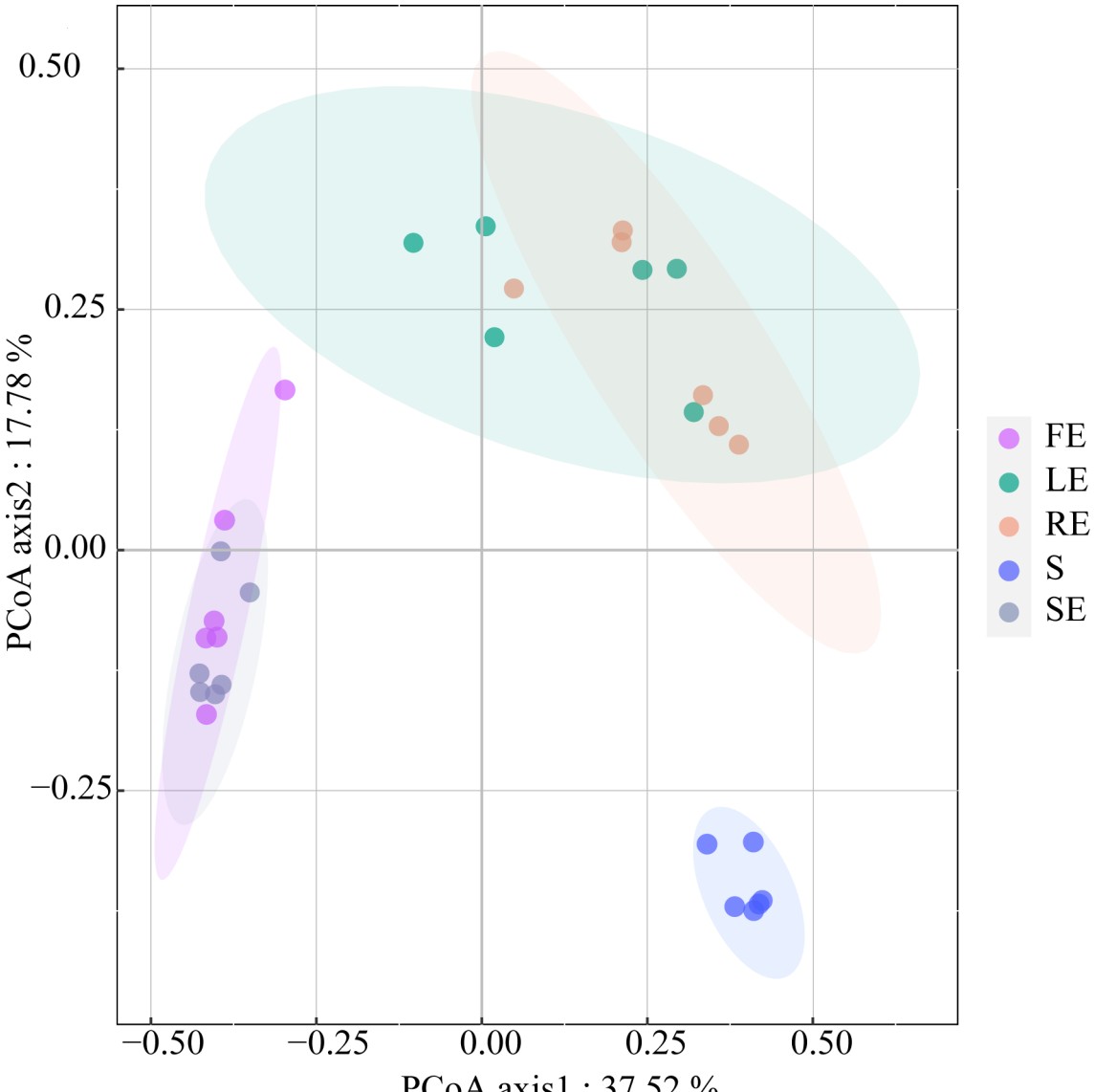

**FIG 2** PCoA based on Bray-Curtis distance showing the bacterial community composition of different ecological niches of *Camellia oleifera*. The FE, LE, SE, RE, and S represent the fruit endosphere, leaf endosphere, stem endosphere, root endosphere, and bulk soil, respectively.

SE, $R^2$ = 0.1711, $P < 0.05$; LE and SE, $R^2$ = 0.4235, $P < 0.01$; RE and SE, $R^2$ = 0.5017, $P < 0.01$) was significantly different.

To understand the influence of niche on the difference in bacterial community composition of *Camellia oleifera*, we decomposed β-diversity into species replacement and richness difference. The results of β-diversity decomposition analysis showed that the difference of bacterial community composition in each niche of *Camellia oleifera* was dominated by the process of species replacement, which contributed 69.90%; however, the impact of the richness difference process on β-diversity is relatively small, which contributed 30.14% (Fig. 3).

## Bacterial community composition in each plant compartment

A total of 1,720 OTUs were generated from all samples, with significant differences among plant compartments (Fig. 4). The numbers of OTUs in the S and root compartments were significantly higher than that in FE, SE, and LE ($P < 0.05$) (Fig. 4A). Compared with FE, SE, and LE, the S increased by 33.20%, 37.03%, and 18.42%, and the RE increased

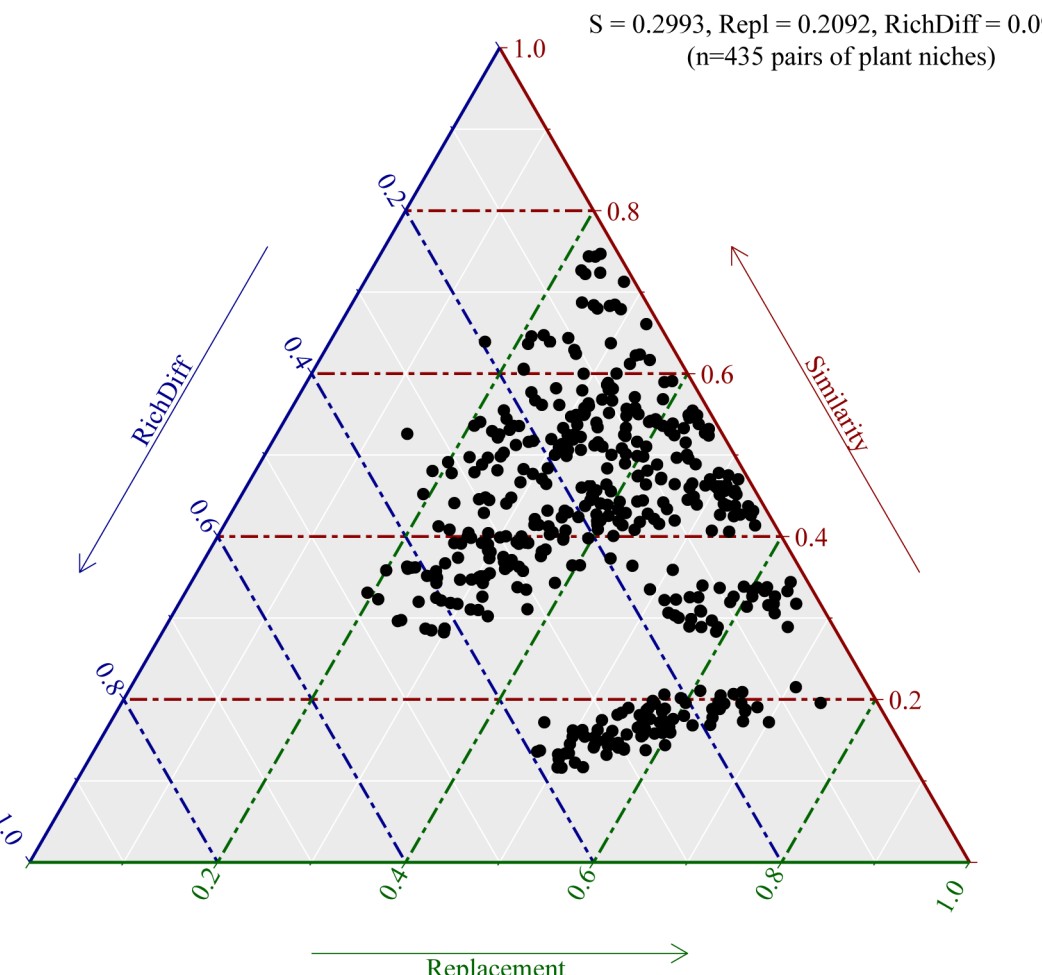

**FIG 3** Triangular plots of beta diversity comparisons (using Sørensen dissimilarity index) for bacterial communities among all samples. Each point represents a pair of niches. Its position is determined by a triplet of values from the S (similarity), Repl (replacement), and RichDiff (richness difference) matrices; each triplet sums to 1. The mean values of S, Repl, and RichDiff are shown.

by 31.72%, 35.63%, and 16.60%, respectively. The OTU number of LE was 18.13% and 22.81% higher than FE and SE, respectively. The shared OTU in different plant compartments of *Camellia oleifera* is shown in Fig. 4B. FE, LE, SE, RE, and S share 537 OTUs; FE, LE, SE, and RE share 325 OTUs; LE, SE, RE, and S share 91 OTUs; FE, LE, RE, and S share 82 OTUs; LE, SE, and S share 245 OTUs; and SE and S share 77 OTU. For other groups, the number of shared OTUs is less than 30. The number of unique OTUs in the plant compartment is as follows: S is 205, RE is 4, and LE is 1.

About 58.32%–90.64% of the 1,720 OTUs obtained are classified at the phylum level. The results of difference analysis showed that 1,269, 833, 1,169, and 860 different OTUs were obtained for FE, LE, SE, and RE, respectively, compared with S (Fig. 5). Among them, 365 OTUs were significantly enriched in at least one compartment. As shown in the "tail" in the MA figure (Fig. 5), LE and RE are similar to S. Compared with depleted OTU, the statistically significant high ratio of enriched OTU suggested the enrichment effect of fruit (795 vs 474). Compared with depleted OTU, the statistically significant high ratio of enriched OTU suggested the enrichment effect of fruit (795 vs 474). By contrast, although the SE enriched many OTUs, it also consumed a larger proportion of OTUs (365 vs 804).

According to the analysis of OTU annotation results, the bacteria in different compartments of *Camellia oleifera* include 307 genera and 26 phyla (Fig. 6). In all samples, the dominant (average relative abundance >1%) phyla are *Proteobacter* (24.76%–78.16%), *Acidobacter* (0.53%–45.06%), *Actinobacteria* (3.78%–13.78%), *Bacteroidetes* (2.19%–

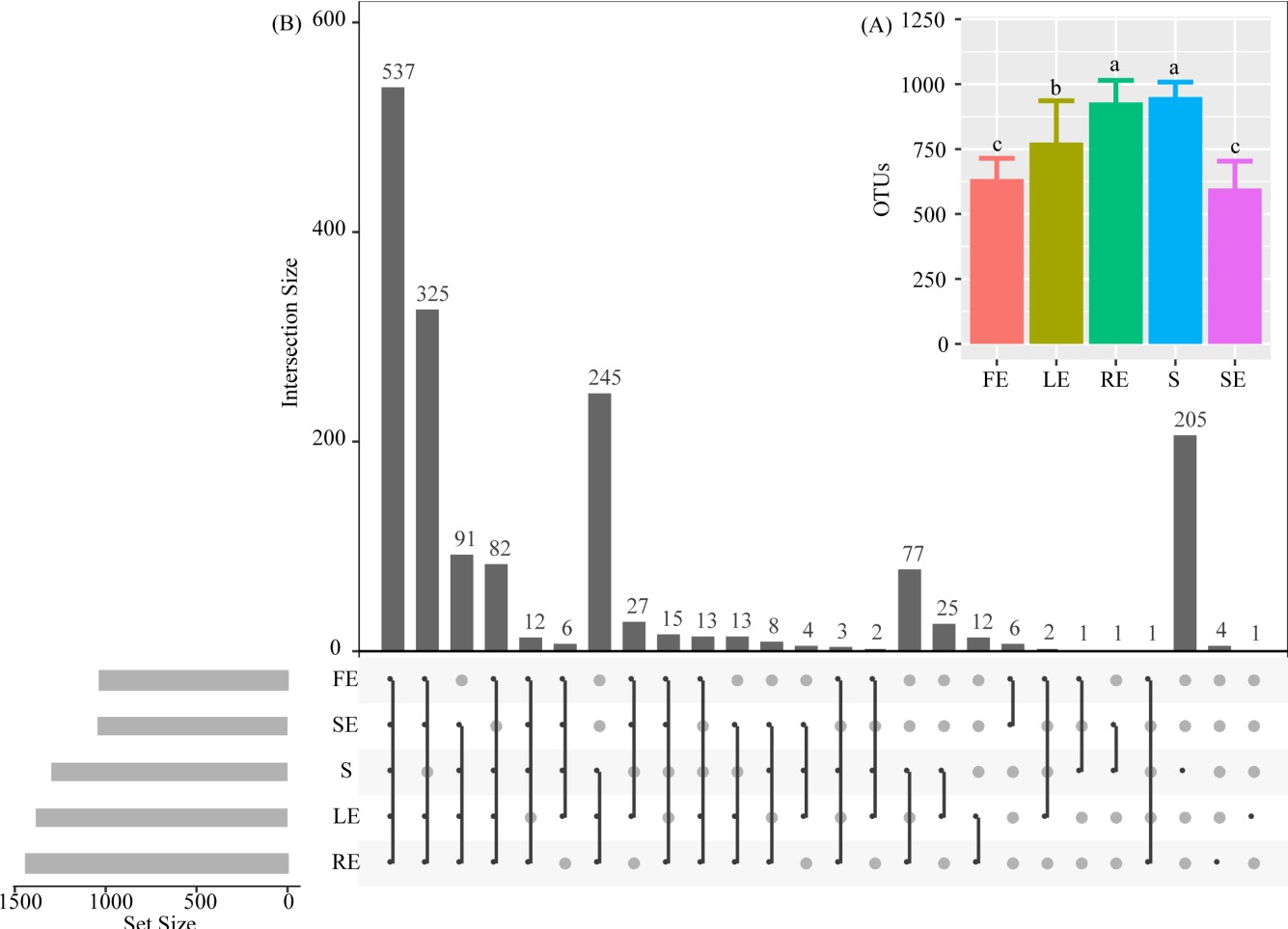

**FIG 4** OTU analysis of the bacterial community in different ecological niches of *Camellia oleifera*. The values represent the mean ± SD (*n* = 6). Different letters indicate significant differences (*P* < 0.05) among each compartment. The FE, LE, SE, RE, and S represent the fruit endosphere, leaf endosphere, stem endosphere, root endosphere, and bulk soil, respectively.

11.28%), *Firmicutes* (0.52%–14.97%), *Chloroflexi* (0.001%–10.83%), and *Verrucomicrobia* (0.01%–7.14%). Dominant genera are *1174–901-12* (0.02%–16.51%), *Sphingomonas* (0.40%–21.59%), *Metallobacterium* (0.01%–12.49%), *Acidothermus* (0.03%–8.33%), *Acidiphilium* (0.02%–8.86%), *Massilia* (0.04%–8.31%), *Hymenobacter* (0.0005%–4.42%), *Candidatus_ Soliactor* (0.02%–4.32%), *Amnibacterium* (0.0002%–4.90%), and *Mucilagini-bacter* (0.10%–3.27%).

Although the top 10 bacteria categories in different compartments are the same, there are significant differences in relative abundance (Fig. 6; Table S2). Take phyla as an example: *Proteobacteria*, *Bacteroidetes*, *Firmicutes*, *Fusobacteria*, and *Spirochaetes* all showed significant enrichment effects of plant compartments (roots, stems, leaves, and fruits). Compared with bulk soil, *Epsilonbacteraeota* (0.09%) is significantly enriched in fruit (*P* < 0.05); *Epsilonbacteraeota* (0.39%), *Gemmatimonadetes* (0.25%), and *Nitrospirae* (0.23%) are enriched (*P* < 0.05) in leaf, *Actinobacteria* (13.78%), *Deferribacteres* (0.33%), *Epsilonbacteraeota* (0.33%), and *Gemmatimonadetes* (0.32%) are enriched (*P* < 0.05) in root. Compared with fruit, *Actinobacteria*, *Acidobacteria*, *Chloroflexi*, *Elusimicrobia*, *Epsilonbacteraeota*, *GAL15*, *Gemmatimonadetes*, *Patescibacteria*, and *WPS-2* are significantly enriched in leaf and root, *Nitrospirae* is significantly enriched in leaf, while *Deferriactors*, *Dependenciae*, and *Verrucomicrobia* are significantly enriched in root samples. Compared with stem, *Chloroflexi*, *Epsilonbacteraeota*, *Firmicutes*, *Fusobacteria*, *Gemmatimonadetes*, and *Planctomycetes* are significantly enriched in leaf and root,

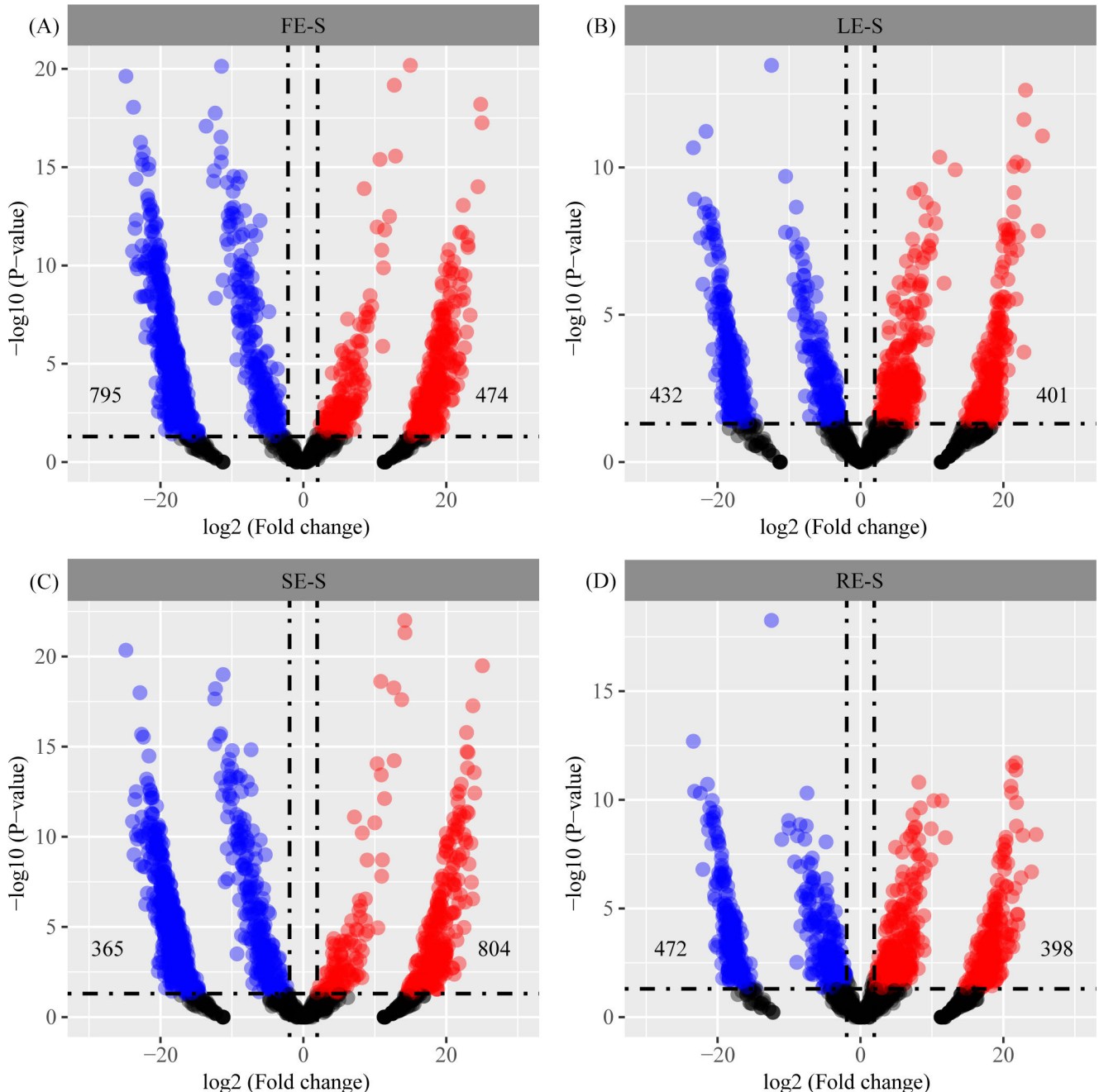

**FIG 5** Enrichment and depletion for each plant compartment of *Camellia oleifera* compared with bulk soil controls as determined by differential abundance analysis. Each point represents an individual OTU, and the position along the *x*-axis represents the abundance fold change compared with bulk soil. The blue dot represents a significant increase in OTU, and the red dot represents a significant decrease.

*Nitrospirae* and *Spirochaetes* were significantly enriched in leaf; *Acidobacteria*, *Deferrichar-acters*, *Dependentiae*, *Elusimacrobia*, and *WPS-2* were significantly enriched in root. Finally, compared with other plant regions, the relative abundance of *Acidobacteria* in fruit (0.53%) is significantly decreased, and *Armatimonadetes* in stem (0.07%) is significantly increased. Table S2 lists the significant impacts of all phyla-spanning compartments.

Linear discriminant analysis (LDA) and effect size analysis (LEfSe) were used for the quantitative analysis of biomarkers (Fig. 7; Fig. S2). We detected significant differences in the abundance of bacterial biomarkers from different tissues and identified a total of 548

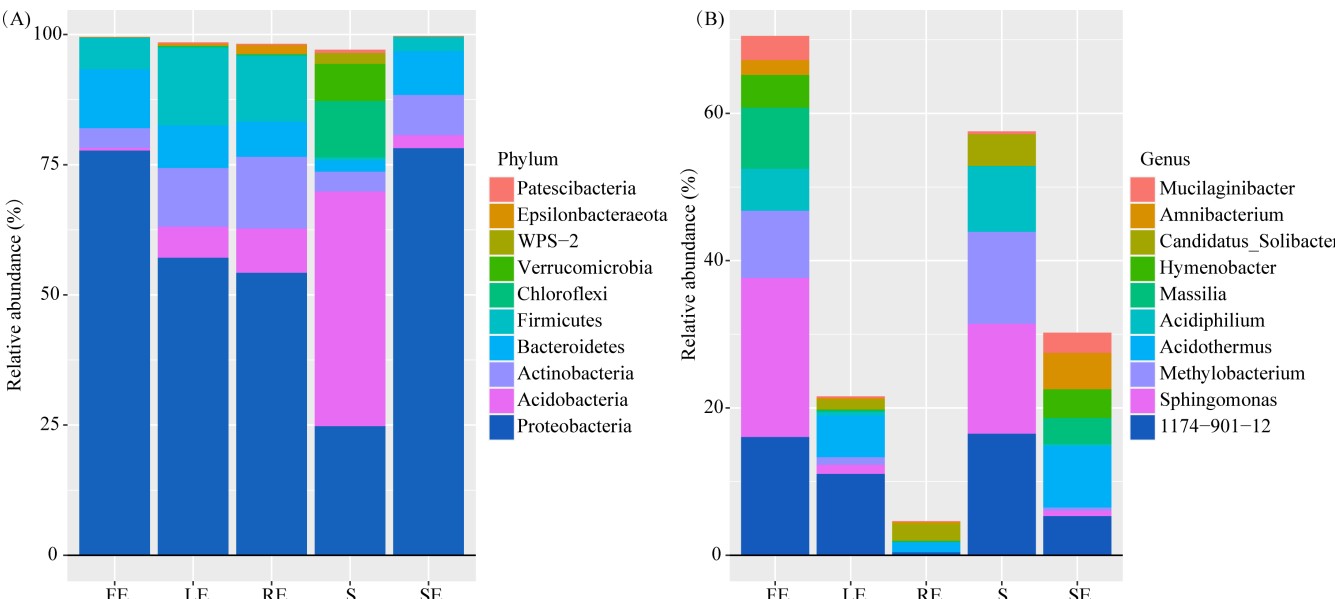

**FIG 6** Top 10 relative abundances of bacterial communities classified at phylum and genus level in different plant compartments of *Camellia oleifera*. The FE, LE, SE, RE, and S represent the fruit endosphere, leaf endosphere, stem endosphere, root endosphere, and bulk soil, respectively.

biomarkers from all samples. As shown in Fig. 7, the important taxa in the fruit belong to *Bacteroidetes* and *Proteobacteria*; the significantly rich taxa in the leaf are *Firmicutes*, *Nitrospirae*, and *Spirochaetes*. Important taxa in root belong to *Actinobacteria*, *Degerribacteres*, *Epsilonbateraeota*, *Fusobacteria*, and *Gemmatimonadetes*. Important taxa in bulk soil samples belong to *Acidobacteria*, *Armatimonadetes*, *Chloroflexi*, and *Verrucomicrobia*.

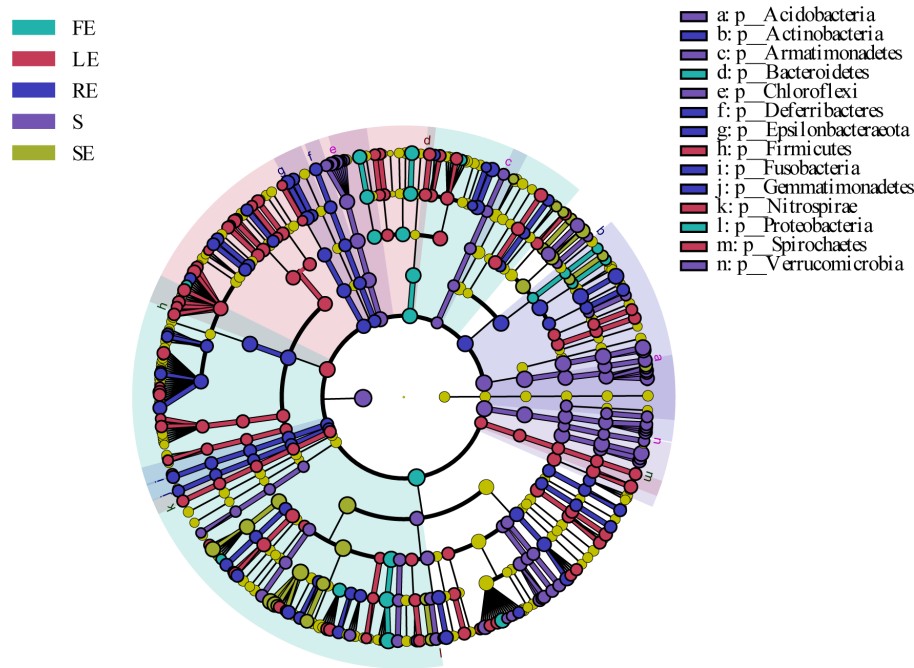

**FIG 7** LEfse analysis of microbial abundance in different plant compartments of *Camellia oleifera*. The histogram of LDA scores computed for differentially abundant bacterial communities among different *Camellia oleifera* compartments identified with a threshold value of 3.0. The FE, LE, SE, RE, and S represent the fruit endosphere, leaf endosphere, stem endosphere, root endosphere, and bulk soil, respectively.

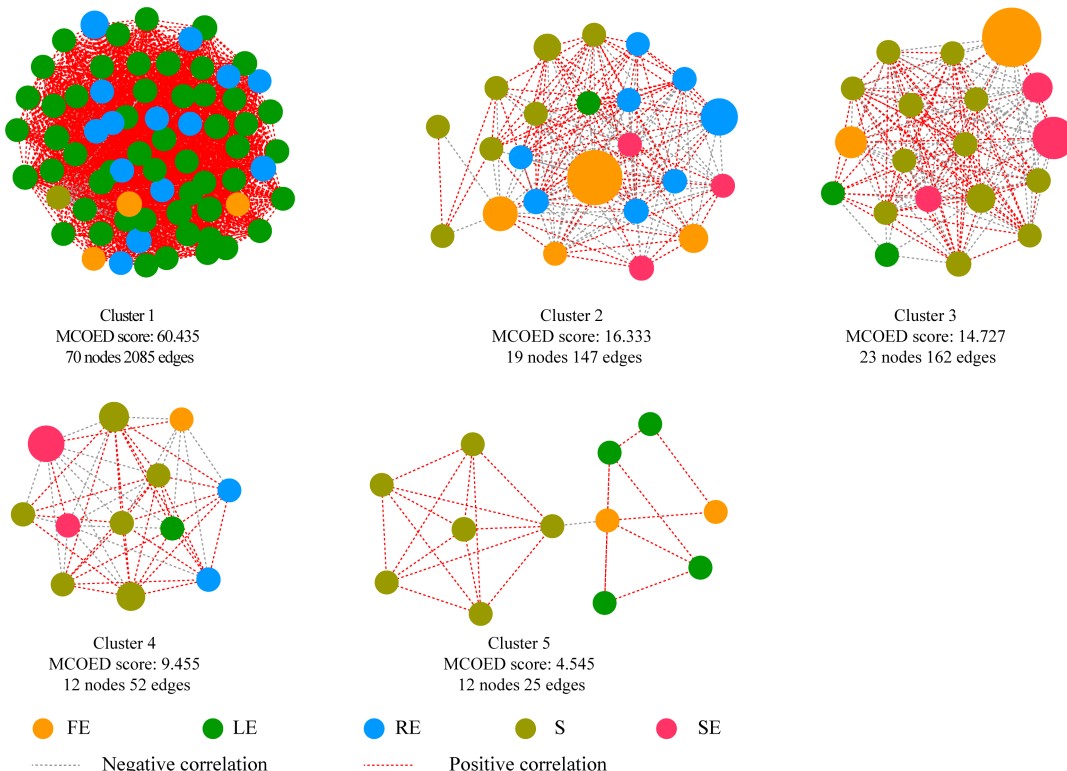

Cluster 1
MCOED score: 60.435
70 nodes 2085 edges

Cluster 2
MCOED score: 16.333
19 nodes 147 edges

Cluster 3
MCOED score: 14.727
23 nodes 162 edges

Cluster 4
MCOED score: 9.455
12 nodes 52 edges

Cluster 5
MCOED score: 4.545
12 nodes 25 edges

● FE   ● LE   ● RE   ● S   ● SE

┄┄┄ Negative correlation       ┄┄┄ Positive correlation

**FIG 8** Bacterial co-expression network diagram of *Camellia oleifera*. Each node represents one genus and an edge is drawn between OTUs if they share a Pearson correlation of greater than or equal to 0.6. The size of the node is proportional to the MCODE score and is color marked at the compartments.

## Identification of hub bacteria

To further explore the interaction between microorganisms in the microenvironment of *Camellia oleifera*, we used the genera with relative abundance (>0.01%) to conduct a co-occurrence network analysis (MCODE: node score cutoff = 0.2, K-core = 2) and visualized the correlation between the genera and each niche of *Camellia oleifera* (Fig. 8). In the bacterial community, we obtained five modules with obvious differences in species characteristics (Fig. 8). Cluster 1 is dominated by leaf endophytic bacteria community, which contains 229 OTUs distributed in 9 phyla and 70 genera, among which *Firmicutes* (5.23%), *Proteobacteria* (3.26%), and *Bacteroides* (2.11%) are the dominant phyla. Cluster 2 is dominated by soil bacterial communities, and 99 OTUs are distributed in 7 phyla and 19 genera, of which *Proteobacteria* (15.18%) is the dominant phyla. In Cluster 3, 176 OTUs belong to 23 genera in five phyla, among which *Proteobacteria* (13.90%) is the dominant phyla.

We use Cytohubba to analyze the sub-network Cluster 1 with the highest score and determine *Nitrospira*, *Haemophilus*, *Staphylococcus*, *Ruminiclostridium,* and *Ochrobactrum* as the core bacterial community of *camellia oleifera* (Table 2).

**TABLE 2** The network topology properties of hub bacteria

| Genus | MCODE score | Degree | Betweenness centrality | Closeness centrality |
|---|---|---|---|---|
| *Nitrospira* | 44.30 | 66 | 0.0012 | 0.8625 |
| *Ruminiclostridium* | 42.60 | 56 | 0.0010 | 0.8415 |
| *Ochrobactrum* | 42.23 | 52 | 0.0006 | 0.8023 |
| *Staphylococcus* | 43.13 | 69 | 0.0036 | 0.8593 |
| *Haemophilus* | 43.21 | 59 | 0.0015 | 0.9170 |

## DISCUSSION

### The diversity of endophytic bacteria community in *Camellia oleifera* is highly dependent on compartment

The results show that the richness and diversity of the *Camellia oleifera* bacterial community gradually decreased from the bulk soil to the endophytic compartment (Fig. 1; Table S1). This result is consistent with the general view of endophyte colonization. This is because the rhizosphere soil-root interface acts as a selective barrier, and only a limited number of bacteria can adapt to the endogenous lifestyle and dominate the endogenous combination, thus forming a unique, highly rich, and diverse microbial community (37). Some studies have found that niche differentiation, especially between soil and plant tissue, can lead to changes in bacterial community structure (38, 39). In this study, there are significant differences in the bacterial community structure in different niches of *Camellia oleifera*, especially between bulk soil and plant tissues (Fig. 2). The species replacement process (69.90% contribution rate) is the dominant factor causing this difference (Fig. 3; Table 1). This result is consistent with the view that each plant compartment is a unique niche of microbial entities and has a unique microbial combination compared with other plant tissues (including roots, stems, leaves, flowers, and seeds) (8, 40, 41).

### Each compartment represents the unique niche of *Camellia oleifera* bacteria

The plant endophytic environment is considered to be a restricted niche. A variety of biological factors (infiltration pathway, plant genotype, strain type, etc.) and abiotic factors (ultraviolet radiation, temperature, dryness, etc.) limit the colonization of endophytic bacteria (7, 42, 43). In this study, the number of OTUs in fruit, leaves, root, and stem was reduced by an average of 33.33% compared to the bulk soil, indicating that it is the main site of microbial colonization and activity in the soil, which harbors a rich and diverse bacterial population as compared to other plant ecological niches (44). In addition, 1,269, 833, 860, and 1,169 differential OTUs were obtained from fruit, leaves, root, and stem, respectively, compared to bulk soil samples (Fig. 5). Contrary to stem (up 365 vs down 804), fruit showed obvious enrichment effect (up 795 vs down 474). This result is consistent with the conclusion that the diversity of endophytic bacterial communities in *Camellia oleifera* is highly dependent on compartment, indicating that only adaptive and non-selective bacterial populations can survive and/or proliferate in the tissues of *Camellia oleifera* (45). Finally, we found that most OTUs are not shared, especially those found in bulk samples (Fig. 5). These findings are consistent with studies showing plant niche differentiation (11, 46). Of these, four OTUs (such as *Ruminococcaceae*, *Chitinophagaceae*, and *Diplorickettsiaceae*) are unique to the root tissue and one (*Methylopila*) to the leaf tissue. In all, 205 OTUs are unique to the soil of *Camellia oleifera* forest, with the majority of them belonging to functional groups that participate in nutrient transformation (*Candidatus_Xiphinematobacte*, *HSB_OF53-F07*, and *FCPS473*), promote plant growth (*Mucilaginibacter*), and decompose organic matter (*Candidatus_Udaeobacter*, *Chthoniobacter*, and *Pedosphaera*) (47–52).

At the species level, *Proteobacteria*, *Actinobacteria*, and *Bacteroidetes* are the common dominant bacteria of *Camellia oleifera*, watermelon, Arabidopsis, rice, Antarctic vascular plants, Dendrobium, and other plants (2, 45, 53–55). This indicates that the composition of endophytic bacterial communities in plants may be similar at the phyla level. It has been found that *Proteobacteria* plays a dominant role in endoderm (56, 57), leaf (45), and stem (11). In this study, *Proteobacteria*, *Bacteroidetes,* and *Firmicutes* were significantly enriched in the fruit, leaf, root, and stem of *Camellia oleifera* (Table S2), and *Proteobacteria* was the dominant phylum (Fig. 6; Table S2). *Sphingomons* are a common bacteria associated with each other in different plant tissues (58), which is significantly enriched in *Camellia oleifera* fruits (21.59%). *Sphingomons* is not only an important regulator of *Arabidopsis thaliana* leaf microbiota (59) but also the most characteristic microorganism in rice seed disease resistance phenotype. It plays the role of "extending the immune

system" in the "disease triangle," and can be passed from generation to generation in the microbiome of healthy plant seeds (60). However, the significance of *sphingomons* expression in *Camellia oleifera* fruits needs further study. *Massilia* can colonize in plant tissues such as *Alopecurus aequalis Sobol* and ryegrass, and degrade polycyclic aromatic hydrocarbons (PAH) compounds (61). In this study, *Massilia* was significantly enriched in *Camellia oleifera* fruits (9.12%) and stems (3.57%), indicating that *Camellia oleifera* may be able to eliminate environmental pollution by degrading PAH through *Massilia*, providing a new perspective for plants to control PAH absorption through endophytic bacteria and reflect the ecological function of *Camellia oleifera* forest. In addition, we also found a variety of bacteria with nitrogen fixation potential in *Camellia oleifera* fruits (*Hymenobacter*, *Allorhizobium-Neorhizobium-Pararhizobium-Rhizobium*), leaves (*Bacillus*, *Mesorhizobium*, *Ochrobacterium*, *Pantooa*, *Pseudomonas*, *Stenotrophomona*, etc.), roots (*Bradyrhizobium*), and stems (*Devosia*) (Fig. S3). The above results indicate that the specific bacteria in the endophytic bacteria of *Camellia oleifera* may play an important role in eliminating environmental pollution and obtaining nutrition.

## Potential ecological functions of hub microbes in *Camellia oleifera*

Hub microbes can be important nodes in the community, rich taxonomic groups in the network structure of the microbial community, or microbes significantly related to ecological functions (62). In our study, five cluster modules were identified in the *Camellia oleifera* bacterial co-occurrence network (Fig. 8). According to the analysis of Cluster 1 with the highest score, the hub bacteria of *Camellia oleifera* are *Nitrospira* (0.07%), *Haemophilus* (0.05%), *Staphylococcus* (0.17%), *Ruminiclostridium* (0.04%), and *Ochrobactrum* (0.14%) (Table 2). *Nitrospira i*s one of the most widely distributed and diverse nitrites oxidizing bacteria, and also a key nitrifying bacteria in natural ecosystems (63, 64). *Ochrobacterium* and *Staphylococcus* can promote the growth of host plants by generating indole-3-acetic acid (IAA) or cooperating with silicate (65, 66). *Ruminiclostridium*, as a cellulose-degrading bacterium, its cellulose-degrading products can provide a carbon source for the growth of other microorganisms on the one hand (67), and may play an indirect role in activating plant defense on the other hand (68). *Haemophilus* is usually related to human pathogens, but it has been found that the genus inhabits plants (69). The hub microbes identified by us are relatively low in abundance, but they are widely colonized in various tissues (fruit, leaf, stem, root, and bulk soil) of *Camellia oleifera*, and may play an important role in nitrogen cycling, host growth promotion, and plant defense.

## Conclusion

In general, in this study, highly diversified and structured niche-specific groups were observed in different sample types of *Camellia oleifera*. The diversity of endophytic bacterial communities in *Camellia oleifera* is highly dependent on plant compartments, and each compartment represents a unique niche of the bacterial community. Our study not only confirms the niche differentiation of the microbes at the soil-root interface but also demonstrates the fine-tuning and adaptation of the endophytic microbiota in the stem, leaf, and fruit compartments. In addition, we have identified the hub bacterial microbes of *Camellia oleifera*. This study provides a relevant model for the systematic study of the changes in microbial community in the organizational level niche of *Camellia oleifera* plants. These results fill the knowledge gap of the endophytic bacterial community of *Camellia oleifera* and provide a theoretical basis for the subsequent exploration of microbial functions and research on bio-fertilizers.

## ACKNOWLEDGMENTS

Yan Zhang, Yang Wu, Hui Wen Zhou, and Li Sha Zhang contributed to the study conception and design. Material preparation, data collection, and analysis were performed by Yan Zhang, Chu Ting Ding, Tao Ya Jiang, and Yu Hua Liu. The first draft

of the manuscript was written by Yan Zhang and all authors commented on previous versions of the manuscript. All authors read and approved the final manuscript.

This work was supported by the National Nature Science Foundation of China (32060010), the Science and Technology Project of Jiangxi Provincial Education Department (GJJ2201935), and the Jiangxi Provincial Natural Science Foundation (20232BAB205052).

## AUTHOR AFFILIATIONS

[1]Institute of Jiangxi Oil-tea Camellia, Jiujiang University, Jiujiang, Jiangxi, China
[2]College of Pharmacy and Life Science, Jiujiang University, Jiujiang City, Jiangxi Province, China

## AUTHOR ORCIDs

Yan Zhang  http://orcid.org/0000-0002-3051-1878
Yang Wu  http://orcid.org/0000-0001-6750-3479

## FUNDING

| Funder | Grant(s) | Author(s) |
|---|---|---|
| The Science and Technology Project of Jiangxi Provincial Education Department | GJJ2201935 | Yan Zhang |

## AUTHOR CONTRIBUTIONS

Yan Zhang, Conceptualization, Funding acquisition, Methodology, Project administration, Software, Writing – original draft, Writing – review and editing | Chu Ting Ding, Data curation, Investigation, Software | Taoya Jiang, Data curation, Investigation, Resources | Yu Hua Liu, Data curation, Investigation, Resources | Yang Wu, Conceptualization, Formal analysis, Methodology, Resources, Software | Hui Wen Zhou, Conceptualization, Formal analysis, Methodology, Resources, Software | Li Sha Zhang, Conceptualization, Formal analysis, Investigation, Methodology, Resources, Software | Ye Chen, Funding acquisition

## DATA AVAILABILITY

The data sets generated during and/or analyzed during the current study are available from the corresponding author upon reasonable request. The raw sequence data reported in this paper have been deposited in the Genome Sequence Archive in BIG Data Center, Beijing Institute of Genomics (BIG), Chinese Academy of Sciences, under accession number CRA009115, which is publicly accessible at https://bigd.big.ac.cn/gsa.

## ADDITIONAL FILES

The following material is available online.

### Supplemental Material

**Figures S1 to S3, Tables S1 and S2 (Spectrum01335-23-S0001.pdf).** Sequencing depth, LDA scores, Faprotax function predictions, α-diversity, enrichment effect.

### Open Peer Review

**PEER REVIEW HISTORY (review-history.pdf).** An accounting of the reviewer comments and feedback.

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
