## [Reviewer comments · Microbiology Spectrum]

Microbiology Spectrum

Community structure and niche differentiation of endosphere bacterial microbiome in *Camellia oleifera*

Yan Zhang, Chuting Ding, Taoya Jiang, Yuhua Liu, Yang Wu, Huiwen Zhou, and Lisha Zhang

Corresponding Author(s): Yan Zhang, Jiujiang University

Review Timeline:

Submission Date:	March 27, 2023
Editorial Decision:	July 19, 2023
Revision Received:	August 30, 2023
Accepted:	September 10, 2023

Editor: Zhongxiong Lai

Reviewer(s): Disclosure of reviewer identity is with reference to reviewer comments included in decision letter(s). The following individuals involved in review of your submission have agreed to reveal their identity: Shi-Hong Zhang (Reviewer #1)

Transaction Report:

DOI: <https://doi.org/10.1128/spectrum.01335-23>

July 19, 2023

Dr. Yan Zhang
Jiujiang University
Institute of Jiangxi Oil-tea Camellia
No.551 Qianjin East Road
Jiujiang, Jiangxi 332005
China

Re: Spectrum01335-23 (Community structure and niche differentiation of endosphere bacterial microbiome in *Camellia oleifera*)

Dear Dr. Yan Zhang:

Link Not Available

Sincerely,

Zhongxiong Lai

Journals Department
Reviewer comments:

Reviewer #1 (Comments for the Author):

Camellia oleifera as a very valuable wooden oil plant. *Camellia oleifera* grown in mountains are facing a significant decrease in soil nutrients and in a low -yield state. The authors evaluated the niche differentiation of bacterial communities of *Camellia oleifera* through 16S rRNA sequencing. The manuscript analyzed the dominant groups of *Camellia oleifera* bacteria, and pointed a large number of beneficial bacteria in *Camellia oleifera*. This study just provides a theoretical basis for the subsequent exploration of microbial functions and research on bio-fertilizers. Considering bio-fertilizer use, the author should conduct relevant research on the isolation and identification of cultivable microorganisms, however, which is clearly overlooked.

Reviewer #2 (Comments for the Author):

The composition and structural characteristics of *Camellia oleifera* microbiome at niche level were systematically studied. Its experiment design, data, and figures behind this paper are good, but the innovative contributions need to be highlighted and strengthened. In the introduction part, there is no detailed information about the impact of plant compartment on bacterial community diversity. For materials and methods, the information for samples name is confusing. In the results, several sentences need to be reorganized with logic and some pictures need to be improved. In the discussion part, the results of some parts are repeated, and their findings need to be explained and discussed in more detail. The scientific question for this manuscript should be highlighted and the English language need further improvement. Therefore, I recommend a major revision for the current version.

1. In the introduction, authors did not explain the connection between the research object and the research content.
2. In the introduction, authors elucidates that the composition of microbial species is influenced by plant compartments. How does the diversity of microorganisms change?
3. The authors told readers it was "bulk soil" in line 73, 311, but is described as "rhizosphere soil" in line 74, 316. This is also the case in other parts of the manuscript.
4. What type of climate is it in the experimental site?
5. Are there any management measures in the *Camellia oleifera* forest?
6. Why not measure the bacteria in other season?
7. Line 162: "had no significant difference, but the diversity index (Simpson: 0.947 {plus minus} 0.02, Shannon: 5.2 {plus minus} 0.8) were significantly reduced." This sentence is not clearly worded! Reword and clarify the numbers.
8. Line 225: The authors defined that the relative abundance > 1% was dominant phyla, while the relative abundance of *Acidobacter* (0.53-45.06%), *Chloroflexi* (0.001-10.83), and *Verrucomimicrobia* (0.01-7.14%) was very small. Contradictory statement claim?
9. Fig.5: The title annotation in Figure 5 should remain consistent with other images.
10. Fig.6B legend is not clear, please modify!
11. Fig.8 is not clear, please modify!
12. The discussion is good in structured. But also have some repetition of results is present, please delete.
13. Line 315: "In addition, our research also shows that most OTUs are not shared". There is a lack of summary, analysis, and discussion on unique OTUs.
14. The whole conclusion are mostly the repetitive of the results, authors need to find new points to address the question.
15. Line346-348: Please improve/rephrase the sentences. "Hub microbes are not necessarily key points in the community, or taxa which are significant amounts of the microbial community network structure, but ecologically relevant hub microbes are likely to be."

Please check the article carefully. Some minor comments are:

1. Line 20: Firmicutes, Chloroflexi and Verrucomicrobia)
2. Line 21-24: Please improve/rephrase the sentences. "A variety of bacteria (*Hymenobacter*, *Allorhizobium-Neorhizobium-Pararhizobium-Rhizobium*, *Mesorhizobium*, *Bradyrhizobium*, *Bacillus*, *Ochrobactrum*, *Pantoea*, *Pseudomonas*, etc.) with a variety of nitrogen -fixed potentials are enriched in *Camellia oleifera* tissue."
3. Line 46-47: "The results of the Xiong et al. (8) pointed out that some members of Burkholderiaceae, Microbacteriaceae, Streptomycetaceae and Rhizobiaceae were enriched on the surface of phylloplane" the names of bacteria are in italics.
4. Line 61-62: "the related endophytic nitrogen fixing bacteria (*Azospirillum brasilense*, *Herbaspirillum seropodicae*, *Burkholderia unamae*)." the names of bacteria are in italics.
5. Line 74: stems, leaves and fruits of *Camellia oleifera*.
6. Line 146 References
7. Line 157-156: Optimize the "The bacterial richness (ACE and Chao1 index) and diversity (Simpson and Shannon index) of the bulk soil (S) of *Camellia oleifera* were significantly higher than those of the fruit endosphere (FE), leaf endosphere (LE) and stem endosphere (SE), and slightly higher than those of the root endosphere (RE)." sentence.
8. Line 199: root chamber?
9. Line 200: "in FE, SE and LE (P < 0.05)" could be changed to "in FE, SE, and LE (P < 0.05)"
10. Line 201: "31.72%, 35.63% and 16.60%," could be changed to "31.72%, 35.63%, and 16.60%,"
11. Line 213: "bulk soil" could be changed to "bulk soil"
12. Line 225: "(3.78-13.78%), Bacteroidetes (2.19-11.28%), Firmicutes (0.52-14.97), Chloroflexi (0.001-10.83) and" could be changed to "(3.78-13.78%), Bacteroidetes (2.19-11.28%), Firmicutes (0.52-14.97%), Chloroflexi (0.001-10.83%)"
13. Line 228: "Massilia (0.04-8.31)" could be changed to "Massilia (0.04-8.31%)"
14. Line 245: "Planctomycotes" could be changed to "Planctomycetes"
15. Line 245: "fruit is significantly decreased (0.53%)" could be changed to "fruit (0.53%) is significantly decreased"
16. Line 268: "relative abundance (>0.01%)" could be changed to "relative abundance (> 0.01%)"
17. Line 304: "bulk soil is 33.20%, 18.42%, 2.18% and 37.03%" could be changed to "bulk soil is 33.20%, 18.42%, 2.18%, and 37.03%"
18. Line 305: "leaf, root and stem, respectively" could be changed to "leaf, root, and stem, respectively"
19. Line 306: "Compared with other plant niches, it contains rich and diverse bacteria (40)." It is really confusing.
20. Line 313, 315, 375: "*camellia oleifera*" could be changed to "*Camellia oleifera*"

21. Line 318: "Proteobacteria, Actinobacteria and Bacteroidetes" could be changed to "Proteobacteria, Actinobacteria, and Bacteroidetes"
22. Line 331: "Arabidopsis thaliana" could be changed to "Arabidopsis thaliana"
23. Line 326: "Alopecurus aequalis Sobol" could be changed to "Alopecurus aequalis Sobol"
24. Reference: 1, 13, 20, 34, 46, 52-56,
25. FIGURE CAPTIONS: "Camellia oleifera" could be changed to "Camellia oleifera"

Staff Comments:

Preparing Revision Guidelines

Please return the manuscript within 60 days; if you cannot complete the modification within this time period, please contact me. If you do not wish to modify the manuscript and prefer to submit it to another journal, please notify me of your decision immediately so that the manuscript may be formally withdrawn from consideration by Microbiology Spectrum.

Dear

Thank you for your letter about our manuscript entitled Community structure and niche differentiation of endosphere bacterial microbiome in *Camellia oleifera* (Spectrum 01335-23). Those comments are all valuable and very helpful for revising and improving our paper, as well as the important guiding significance to our researches. We have studied comments carefully and have made correction. We hope to get approval to re-submit. Revised portion can be seen in the manuscript with tracked changes. The main corrections in the paper and the responds to the reviewer's comments are as flowing:

Comments from the reviewers:

-Reviewer 1

Comments

Camellia oleifera as a very valuable wooden oil plant. *Camellia oleifera* grown in mountains are facing a significant decrease in soil nutrients and in a low -yield state. The authors evaluated the niche differentiation of bacterial communities of *Camellia oleifera* through 16S rRNA sequencing. The manuscript analyzed the dominant groups of *Camellia oleifera* bacteria, and pointed a large number of beneficial bacteria in *Camellia oleifera*. This study just provides a theoretical basis for the subsequent exploration of microbial functions and research on bio-fertilizers. Considering bio-fertilizer use, the author should conduct relevant research on the isolation and identification of cultivable microorganisms, however, which is clearly overlooked.

Response: Many thanks for your interest and approval of our manuscript. For the questions raised by the reviewers, we will improve them in the next step of work. The isolation and identification of cultivable microorganisms is a necessary step in the applied research of microorganisms. Next, we will isolate and identify nitrogen fixing microorganisms from different parts of *Camellia oleifera*, and verify their growth promoting effects.

-Reviewer 2

Comments:

1. In the introduction, authors did not explain the connection between the research object and the research content.

Response: This suggestion is really useful for us to improve our manuscript. According to reviewer's suggestions, we have modified this section. Please see from Line 67-71. As below:

Therefore, analysing the characteristics of bacterial communities in different ecological niches of healthy hosts may help to improve soil quality, crop growth and stress resistance, thus reducing the dependence on fertilisers in production activities. It is of great importance for promoting the sustainable development of *Camellia oleifera* production and understanding the contribution of *Camellia oleifera* to ecosystem services.

2. In the introduction, authors elucidates that the composition of microbial species is influenced by plant compartments. How does the diversity of microorganisms change?

Response: This suggestion is really useful for us to improve our manuscript. We have modified this section. Please see from Line 50-52. As below:

In addition, only adaptive or non-picky bacterial populations can survive or flourish within plant tissues due to filtration and selection, which leads to a low degree of microbial diversity (14).

3. The authors told readers it was "bulk soil" in line 73, 311, but is described as "rhizosphere soil" in line 74, 316. This is also the case in other parts of the manuscript.

Response: We are very sorry for our negligence making reviewers confused. After careful consideration, there is still a significant difference between the rhizosphere soil and the bulk soil. Our rhizosphere soil or bulk soil samples are located approximately 30-50 cm away from the tree and do not contain any roots. We have redefined it as bulk soil. Please see from Line 32, 78, 90, 91, 297, 329.

4. What type of climate is it in the experimental site?

Response: Thanks for your question. Yongxiu County belongs to the transitional zone between the

central and northern subtropics, with a humid monsoon climate. The region is rich in light and heat, with a warm climate and distinct four seasons, making it a suitable planting area for *Camellia oleifera*.

5. Are there any management measures in the *Camellia oleifera* forest?

Response: Thanks for your suggestion. In the planting period of *Camellia oleifera*, no other management measures are taken except for weeding the fields manually each autumn. Please see from Line 100-101. As below:

Weeds were controlled manually in each autumn. No obvious diseases and pest damages were observed in recent years. Irrigation was not applied throughout plant period.

6. Why not measure the bacteria in other season?

Response: Thanks for your suggestion. Spring is a period of vigorous life activity for *Camellia oleifera*. This includes sprouting new buds, roots, and the rapid expansion of fruits. As a result, we have prioritized this period for bacterial testing. In future work, we plan to perform further analysis of microorganisms during different seasons and stages of fruits.

7. Line 162: "had no significant difference, but the diversity index (Simpson: 0.947 ± 0.02 , Shannon: 5.2 ± 0.8) were significantly reduced." This sentence is not clearly worded! Reword and clarify the numbers.

Response: Very sorry for our negligence make reviewer's confusion. As the reviewer's suggestion that we have corrected "Compared with SE, the ACE and Chao1 indexes in FE had no significant difference, but the diversity index (Simpson: 0.947 ± 0.02 , Shannon: 5.2 ± 0.8) were significantly reduced." to "Compared with SE, the diversity index (Simpson: 0.947 ± 0.02 , Shannon: 5.82 ± 0.8) were significantly reduced under FE, but the richness (ACE and Chao1 index) were no significant differences between SE and FE." Please see from Line 166-169.

8. Line 225: The authors defined that the relative abundance > 1% was dominant phyla, while the relative abundance of *Acidobacter* (0.53-45.06%), *Chloroflexi* (0.001-10.83), and *Verrucomimicrobia* (0.01-7.14%) was very small. Contradictory statement claim?

Response: Thanks for your suggestion. This is because different bacteria have different colonization environments in different ecological niches. For example, the relative abundance of *Firmicutes* in LE is 14.97%, while in S it is 0.52%. Therefore, 1% abundance refers to the average value among all samples. Please see from Line 230-231.

9. Fig.5: The title annotation in Figure 5 should remain consistent with other images.

Response: Thanks for your suggestion. As the reviewer's suggestion that we modified Fig.5 in our revised manuscript.

10. Fig.6B legend is not clear, please modify!

Response: Thanks for your suggestion. As the reviewer's suggestion that we modified Fig.6 in our revised manuscript.

11. Fig.8 is not clear, please modify!

Response: Thanks for your suggestion. As the reviewer’s suggestion that we modified Fig.8 in our revised manuscript.

12. The discussion is good in structured. But also have some repetition of results is present, please delete.

Response: Thanks for your suggestion. As the reviewer’s suggestion that we modified “discussion” in our revised manuscript. Please see from Line 310-314, 316-318, 330-336, 365-366.

13. Line 315: "In addition, our research also shows that most OTUs are not shared". There is a lack of summary, analysis, and discussion on unique OTUs.

Response: Thanks for your suggestion. As the reviewer's suggestion that we modified "discussion" in our revised manuscript. Please see from Line 330-336. As below:

In addition, Four OTUs (such as *Ruminococcaceae*, *Chitinophagaceae*, and *Diploricettsiaceae*) are unique to the root tissue and one (*Methylopila*) to the leaf tissue. 205 OTUs are unique to the soil of *Camellia oleifera* forest, with the majority of them belonging to functional groups that participate in nutrient transformation (*Candidatus_Xiphinematobacte*, *HSB_OF53-F07*, and *FCPS473*), promote plant growth (*Mucilaginibacter*), and decompose organic matter (*Candidatus_Udaeobacter*, *Chthoniobacter*, and *Pedosphaera*) (47-52).

14. The whole conclusion are mostly the repetitive of the results, authors need to find new points to address the question.

Response: Thanks for your suggestion. As the reviewer's suggestion that we modified "Conclusion" in our revised manuscript. Please see from Line 385-390. As below:

In general, in this study, highly diversified and structured niche specific groups were observed in different sample types of *Camellia oleifera*. The diversity of endophytic bacterial communities in *Camellia oleifera* is highly dependent on plant compartments, and each compartment represents a unique niche of the bacterial community. Our study not only confirms the niche differentiation of the microbes at the soil-root interface, but also demonstrates the fine-tuning and adaptation of the endophytic microbes in the stem, leaf, and fruit compartment. In addition, the existence of a large number of beneficial bacteria indicates that *Camellia oleifera* plays an important role in eliminating environmental pollution and nitrogen cycling. Finally, we have identified the hub bacterial microbes of *Camellia oleifera*. This study provides a relevant model for the systematic study of the changes of microbial community in the organizational level niche of *Camellia oleifera* plants. These results fill the knowledge gap of the endophytic bacterial community of *Camellia oleifera*, and provide a theoretical basis for the subsequent exploration of microbial functions and research on bio-fertilizers.

15. Line 346-348: Please improve/rephrase the sentences. "Hub microbes are not necessarily key points in the community, or taxa which are significant amounts of the microbial community network structure, but ecologically relevant hub microbes are likely to be."

Response: Following reviewer's suggested that we have corrected the sentence in our revised manuscript. Please see from Line 365-366. As follow:

Hub microbes can be important nodes in the community, rich taxonomic groups in the network structure of the microbial community, or microbes significantly related to ecological functions.

Some minor comments are:

1. Line 20: Firmicutes, Chloroflexi and Verrucomicrobia)

Response: Very sorry for our careless. As the reviewer's suggestion that we have corrected Firmicutes, Chloroflexi and Verrucomicrobia) to "Firmicutes, Chloroflexi, and Verrucomicrobia)" in our revised manuscript. Please see from Line 19.

2. Line 21-24: Please improve/rephrase the sentences. "A variety of bacteria (Hymenobacter, Allorhizobium-Neorhizobium-Pararhizobium-Rhizobium, Mesorhizobium, Bradyrhizobium, Bacillus, Ochrobactrum, Pantoea, Pseudomonas, etc.) with a variety of nitrogen -fixed potentials are enriched in *Camellia oleifera* tissue."

Response: Very sorry for our careless. As the reviewer's suggestion that we have corrected the sentence in our revised manuscript. Please see from Line 22-23. As below:

A variety of bacteria (Hymenobacter, Allorhizobium-Neorhizobium-Pararhizobium-Rhizobium, Mesorhizobium, Bradyrhizobium, Bacillus, Ochrobactrum, Pantoea, Pseudomonas, etc.) with nitrogen -fixed potentials are enriched in *Camellia oleifera* tissue.

3. Line 46-47: "The results of the Xiong et al. (8) pointed out that some members of Burkholderiaceae, Microbacteriaceae, Streptomycetaceae and Rhizobiaceae were enriched on the surface of phylloplane" the names of bacteria are in italics.

Response: Following reviewer's suggested that we have corrected the sentence in our revised manuscript. Please see from Line 45-46. As below:

The results of the Xiong et al. (8) pointed out that some members of *Burkholderiaceae*,

Microbacteriaceae, *Streptomycetaceae*, and *Rhizobiaceae* were enriched on the surface of phylloplane...

4. Line 61-62: "the related endophytic nitrogen fixing bacteria (*Azospirillum brasilense*, *Herbaspirillum seropodicae*, *Burkholderia unamae*)." the names of bacteria are in italics.

Response: Very sorry. As the reviewer's suggestion that we have corrected the sentence in our revised manuscript. Please see from Line 61-62. As below:

...the related endophytic nitrogen fixing bacteria (*Azospirillum brasilense*, *Herbaspirillum seropodicae*, *Burkholderia unamae*).

5. Line 74: stems, leaves and fruits of *Camellia oleifera*.

Response: Very sorry for our careless. As the reviewer's suggestion that we have corrected the sentence in our revised manuscript. Please see from Line 78.

6. Line 146 References

Response: Sorry. As the reviewer's suggestion that we have added the sentence "reference 34" in our revised manuscript. Please see from Line 149. As below:

34. Shen CC, Gunina A, Luo Y, Wang JJ, He JZ, Kuzyakov Y, Hemp A, Classen AT, Ge Y. 2020. Contrasting patterns and drivers of soil bacterial and fungal diversity across a mountain gradient. *Environ Microbiol* 22: 3287-3301. <https://doi.org/10.1111/1462-2920.15090>

7. Line 157-156: Optimize the "The bacterial richness (ACE and Chao1 index) and diversity (Simpson and Shannon index) of the bulk soil (S) of *Camellia oleifera* were significantly higher than those of the fruit endosphere (FE), leaf endosphere (LE) and stem endosphere (SE), and slightly higher than those of the root endosphere (RE)." sentence.

Response: Very sorry. As the reviewer's suggestion that we have corrected the sentence in our revised manuscript. Please see from Line 160-162. As below:

The bacterial richness (ACE and Chao1 index) and diversity (Simpson and Shannon index) of bulk soil (S) were significantly higher than fruit endosphere (FE), leaf endosphere (LE), and stem endosphere (SE), and slightly higher than root endosphere (RE).

8. Line 199: root chamber?

Response: Following the reviewer's comment, we have corrected "root chamber" to "root compartment" in our revised manuscript. Please see from Line 206, 297, 326.

9. Line 200: "in FE, SE and LE ($P < 0.05$)" could be changed to "in FE, SE, and LE ($P < 0.05$)"

Response: Very sorry for our careless. As the reviewer's suggestion that we have corrected "in FE, SE and LE ($P < 0.05$)" to "in FE, SE, and LE ($P < 0.05$)" in our revised manuscript. Please see from Line 207.

10. Line 201: "31.72%, 35.63% and 16.60%," could be changed to "31.72%, 35.63%, and 16.60%,"

Response: Very sorry for our careless. As the reviewer's suggestion that we have corrected corrected "31.72%, 35.63% and 16.60%," to "31.72%, 35.63%, and 16.60%," in our revised manuscript. Please see from Line 208.

11. Line 213: "bulk soil" could be changed to "bulk soil"

Response: Thank you for reviewer's comment. As the reviewer's suggestion that we have corrected "bulk soil" to "S" in our revised manuscript. Please see from Line 220.

12. Line 225: "(3.78-13.78%), Bacteroidetes (2.19-11.28%), Firmicutes (0.52-14.97), Chloroflexi (0.001-10.83) and" could be changed to "(3.78-13.78%), Bacteroidetes (2.19-11.28%), Firmicutes (0.52-14.97%), Chloroflexi (0.001-10.83%)"

Response: Thank you for reviewer's comment that we have changed " (3.78-13.78%), *Bacteroidetes* (2.19-11.28%), *Firmicutes* (0.52-14.97), *Chloroflexi* (0.001-10.83) and" to "(3.78-13.78%), *Bacteroidetes* (2.19-11.28%), *Firmicutes* (0.52-14.97%), *Chloroflexi* (0.001-10.83%), and ..." in our revised manuscript. Please see from Line 232-233.

13. Line 228: "Massilia (0.04-8.31)" could be changed to "Massilia (0.04-8.31%)"

Response: Thank you. As the reviewer's suggestion that we have corrected the sentence in our

revised manuscript. Please see from Line 235. As below:

14. Line 245: "Planctomycotes" could be changed to "Planctomycetes"

Response: Thank you. As the reviewer's suggestion that we have corrected "Planctomycotes" to "*Planctomycetes*" in our revised manuscript. Please see from Line 252.

15. Line 245: "fruit is significantly decreased (0.53%)" could be changed to "fruit (0.53%) is significantly decreased"

Response: Very sorry for our careless. As the reviewer's suggestion that we have changed "fruit is significantly decreased (0.53%)" to "fruit (0.53%) is significantly decreased" in our revised manuscript. Please see from Line 255-256.

16. Line 268: "relative abundance (>0.01%)" could be changed to "relative abundance (> 0.01%)"

Response: Following the reviewer's comment, we have corrected "relative abundance (>0.01%)" to "relative abundance (> 0.01%)" sentence in our revised manuscript. Please see from Line 275.

17. Line 304: "bulk soil is 33.20%, 18.42%, 2.18% and 37.03%" could be changed to "bulk soil is 33.20%, 18.42%, 2.18%, and 37.03%"

Response: Following the reviewer's comment, we have corrected the sentence in our revised manuscript. Please see from Line 312-314.

In this study, the number of OTUs in fruit, leave, root, and stem was reduced by an average of 33.33% compared to the bulk soil.

18. Line 305: "leaf, root and stem, respectively" could be changed to "leaf, root, and stem, respectively"

Response: Sorry for our careless. As the reviewer's suggestion that we have corrected the sentence in our revised manuscript. Please see from Line 312-314.

In this study, the number of OTUs in fruit, leave, root, and stem was reduced by an average of 33.33% compared to the bulk soil.

19. Line 306: "Compared with other plant niches, it contains rich and diverse bacteria (40)." It is really confusing.

Response: Following the reviewer's comment. "it" represents "bulk soil". To avoid confusion among readers, we have changed in our revised manuscript. Please see from Line 316-318.
...indicating that it is the main site of microbial colonisation and activity in the soil, which harbours a rich and diverse bacterial population as compared to other plant ecological niches(44).

20. Line 313, 315, 375: "camellia oleifera" could be changed to "Camellia oleifera"

Response: Very sorry for our careless. As the reviewer's suggestion that we have corrected "*camellia oleifera*" to "*Camellia oleifera*" in our revised manuscript. Please see from Line 326, 328, 401.

21. Line 318: "Proteobacteria, Actinobacteria and Bacteroidetes" could be changed to "Proteobacteria, Actinobacteria, and Bacteroidetes"

Response: Following the reviewer's comment, we have corrected corrected "*Proteobacteria, Actinobacteria and Bacteroidetes*" to "*Proteobacteria, Actinobacteria, and Bacteroidetes*" in our revised manuscript. Please see from Line 337.

22. Line 331: "Arabidopsis thaliana" could be changed to "Arabidopsis thaliana"

Response: Thank you. As the reviewer's suggestion that we have changed "Arabidopsis thaliana" changed "*Arabidopsis thaliana*" in our revised manuscript. Please see from Line 346.

23. Line 326: "Alopecurus aequalis Sobol" could be changed to "Alopecurus aequalis Sobol"

Response: Very sorry for our careless. As the reviewer's suggestion that we have changed "*Alopecurus aequalis Sobol*" changed "*Alopecurus aequalis Sobol* " in our revised manuscript. Please see from Line 350.

24. Reference: 1, 13, 20, 34, 46, 52-56,

Response: As the reviewer's suggestion that we have corrected the sentence in our revised

manuscript. Please see from Line 405-407, 442-445, 469-473, 574-577, 593, 595, 599, 602, 605.

25. FIGURE CAPTIONS: "Camellia oleifera" could be changed to "Camellia oleifera"

Response: Very sorry for our careless. As the reviewer's suggestion that we have corrected the sentence in our revised manuscript. Please see from Line 642, 648.

Other modifications:

1. Change the “NaClO (2.5% active Cl – 105 and 0.1%..” to “NaClO (2.5% active Cl⁻ 105 and 0.1%” . **Line 105.**
2. Change the “the flora” to “compartment” . **Line 326.**
3. Change the “nitrite oxidizing bacteria” to “nitrites oxidizing bacteria”. **Line 373.**
4. Improvements have been made to the formatting of tables. **Table 2, S1, S2.**
5. Added a note to the title of Figure 8. **Line 685.**

September 5, 2023

Dr. Yan Zhang
Jiujiang University
Institute of Jiangxi Oil-tea Camellia
No.551 Qianjin East Road
Jiujiang, Jiangxi 332005
China

Re: Spectrum01335-23R1 (Community structure and niche differentiation of endosphere bacterial microbiome in *Camellia oleifera*)

Dear Dr. Yan Zhang:

My decision is accept.

Your manuscript has been accepted, and I am forwarding it to the ASM Journals Department for publication. You will be notified when your proofs are ready to be viewed.

Sincerely,

Zhongxiong Lai
Editor, Microbiology Spectrum
